# Involving the public in epidemiological public health research: a qualitative study of public and stakeholder involvement in evaluation of a population-wide natural policy experiment

Rachel Anderson de Cuevas,[1] Lotta Nylén,[2] Bo Burström,[2] Margaret Whitehead[1]

## ABSTRACT

**Background** Public involvement in research is considered good practice by European funders; however, evidence of its research impact is sparse, particularly in relation to large-scale epidemiological research.

**Objectives** To explore what difference public and stakeholder involvement made to the interpretation of findings from an evaluation of a natural policy experiment to influence the wider social determinants of health: 'Flexicurity'.

**Setting** Stockholm County, Sweden.

**Participants** Members of the public from different occupational groups represented by blue-collar and white-collar trade union representatives. Also, members of three stakeholder groups: the Swedish national employment agency; an employers' association and politicians sitting on a national labour market committee. Total: 17 participants.

**Methods** Qualitative study of process and outcomes of public and stakeholder participation in four focused workshops on the interpretation of initial findings from the flexicurity evaluation.

**Outcome measures** New insights from participants benefiting the interpretation of our research findings or conceptualisation of future research.

**Results** Participants sensed more drastic and nuanced change in the Swedish welfare system over recent decades than was evident from our literature reviews and policy analysis. They also elaborated hidden developments in the Swedish labour market that were increasingly leading to 'insiders' and 'outsiders', with differing experiences and consequences for financial and job security. Their explanation of the differential effects of the various collective agreements for different occupational groups was new and raised further potential research questions. Their first-hand experience provided new insights into how changes to the social protection system were contributing to the increasing trends in poverty among unemployed people with limiting long-standing illness. The politicians provided further reasoning behind some of the policy changes and their intended and unintended consequences. These insights fed into subsequent reporting of the flexicurity evaluation results, as well as the conceptualisation of new research that could be pursued in a future programme.

## Strengths and limitations of this study

► To our knowledge, this is the first report of the impact of involving the public and stakeholders in population-wide epidemiological studies, which require interpretation of national quantitative datasets.

► It provides a reasoned solution to the challenge of developing the most appropriate public and stakeholder involvement for quantitative public health research on the wider social determinants of health.

► A limitation is that the public and stakeholders were involved in one discrete phase of the research process, rather than throughout.

► This limitation is being addressed by the continued involvement of participants in the conceptualisation and planning of future research from their ideas coming out of the current study.

## INTRODUCTION AND BACKGROUND

Public and patient involvement in health and social research is regarded by funding bodies and policy makers as an integral part of good practice and democratisation of the research process.[1] There is growing recognition that lay people have valuable expertise that should inform the research process, as well as having the right to be more actively involved in the design and outputs of research that is about and for them. Increasingly, research funding bodies also require researchers to demonstrate the value of patient and public involvement by evidencing its impact.[2]

There is now a substantial body of evidence on good practice in involving patients in research to improve the services that they need[3–5] and a growing literature on involving the wider public in community-based prevention and health promotion research, for example, engaging residents of areas in which neighbourhood initiatives are based.[6 7] There are few studies, however,

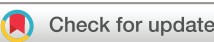

[1]Department of Public Health and Policy, University of Liverpool, Liverpool, UK
[2]Department of Public Health Sciences, Karolinska Institute, Stockholm, Sweden

**Correspondence to**
Prof Bo Burström;
bo.burstrom@ki.se

**BMJ**

about the involvement of the public in population-wide epidemiological studies, including the evaluation of nationwide policies to influence the wider social determinants of health (such as employment and poverty). Arguably, one of the main reasons for this dearth of public involvement studies at the population level is the many additional challenges that public health research of this nature throws up. Not least among these challenges is the selection of the most appropriate representatives from among 'the general public' and the question of where in the process, and in what form, that involvement would be both effective and acceptable to those involved, given the highly technical nature of the statistical analysis of epidemiological datasets.

In this paper, we aim to present the findings of our attempts to tackle some of these challenges, by incorporating public and stakeholder involvement in the evaluation of the impact on health inequalities of a natural policy experiment in labour market 'flexicurity'. We explain what we did and why, and how that involvement benefited the interpretation of our research findings and conceptualisation of future research.

### 'Flexicurity' and the employment of people with a disability or chronic illness

There is a general rise across Europe in the numbers of disabled and chronically ill people of working age that are not in employment. This is a matter of growing concern for governments due to its adverse consequences for public health and potential to exacerbate health inequalities, as well as its effect on increasing social security expenditure.[8] Ageing populations and a raised retirement age in several countries mean that the proportion of the working age population leaving the labour force due to ill-health is expected to rise further.[8 9]

In Sweden, the proportion of the population receiving sickness or activity compensation has trebled in the last 30 years.[10] The combined cost of sick leave and disability pensions equates to approximately 4% of Swedish gross domestic product (GDP) and nearly 8% of the working age population receive disability pensions.[8] The situation in Denmark and Norway is similar (8% and 10%, respectively), while in the UK, more than 2.6 million people receive incapacity-related benefits, accounting for a quarter of the social security budget and 1.5% of GDP.[11]

Being outside the labour market is likely to add to the social and economic exclusion of people with a disability or illness, as income, opportunities to participate fully in society and quality of life decrease.[8 9] People in poor health have lower employment prospects than the general population, and these prospects diminish in line with declining socioeconomic position.[12 13] The social gradient in the employment of people with disabilities may serve to widen health inequalities, since the more disadvantaged who already have a higher prevalence of ill health will be further disadvantaged by the negative effects of unemployment.[9]

European countries have undertaken a range of labour market initiatives that may affect the ability of chronically ill and disabled people to get and keep a job.[8] Population-wide policies centre around two axes: how flexible the labour market is (the degree of regulation and employment rights) and how much financial security the welfare system offers to disabled people who are out of work. These policies have generated considerable debate about their possible impact on the employment opportunities of chronically ill and disabled people in different socioeconomic groups.

Increased flexibility of employment conditions is considered to have contributed to higher levels of job insecurity and to have disproportionately affected those who find themselves in the most vulnerable positions in the labour force (including women, the low skilled and people with disabilities).[14] Since labour market deregulation is considered an essential strategy to redress the economic downturn, 'flexicurity' has been proposed as an alternative, compensating for increased flexibility with provisions of security.[10] Flexicurity policies aim to integrate a flexible labour market with strong financial security and active labour market policies. (Denmark and the Netherlands are often cited as exemplars of flexicurity in its most developed form.) As part of the developing methodologies to reduce inequalities in the determinants of health (DEMETRIQ) project funded by an FP7 EU grant, we investigated the impact of flexicurity policies on social inequalities in the chances of being in employment for people with and without limiting long-standing illness (LLI) in European countries. To do this, we exploited, for research purposes, the contrasts in flexibility and financial security found in exemplar countries, which provided the necessary variation for a natural policy experiment (see, eg, ref 15). The case study in public involvement reported here was conducted as a component of the Swedish country study.

### Study design and methods

The qualitative study aimed to explore how public and stakeholder involvement could be incorporated effectively into the evaluation of the health inequalities impact of flexicurity policies and how that affected (if at all) the interpretation of findings and implications for future research.

### Study participants

The first question to be resolved is 'who is "the public?"' in the context of the evaluation of population-wide employment policies. As the umbrella project was focused on inequalities in health, the EU commissioning brief identified an additional need for 'further research on involving population groups with least power and resources, who are intended to be the ultimate beneficiaries of the research'.[16] With this proviso and the population level context in mind, we made the following selection decisions:

1. The ultimate beneficiaries of our flexicurity research were people in the most vulnerable positions in the labour market, specifically groups with low skills/low education. Those with low education and disabilities/ LLI were doubly disadvantaged.

2. Participants for this explorative qualitative study would be drawn from one European country: Sweden, from where the overall leadership of the flexicurity evaluation was based and which was representative of a recognised type of flexibility and security policy system (ie, low flexibility and high security).

3. We needed to avoid 'tokenism', whereby a few low-skilled workers or unemployed people with disabilities were selected at random, without consideration of the nature of the research data they would be engaging with. Following discussions with Patient and Public Involvement (PPI) experts and deep knowledge of the health inequalities literature, we decided that it would be more appropriate to involve trade union representatives from blue-collar and white-collar unions as members of the relevant public. They were workers themselves but also had the added experience of how their fellow workers in different socioeconomic circumstances and with different health conditions fared under the various national policies. They were also familiar with considering labour market trends at the population level. In addition, we decided to invite participation from representatives of other relevant public bodies, who could be more accurately termed 'stakeholder groups' rather than 'members of the public'. These stakeholders were members of: the Swedish employers' association (as the Swedish labour market is reliant to a greater extent than elsewhere on collective agreements between trade unions and employers); the Swedish national employment agency (as the public body that implements policies to get unemployed people back to work and has a special remit for disabled and chronically ill workers); and politicians from the ruling coalition and opposition parties (who sat on a national labour market committee).

## Study design and conduct

The study component reported here comprised qualitative research involving members of the public (as defined above) and stakeholder groups in the interpretation of research findings and reflections on future research. Four focused workshops with participants were held using a process developed by Petticrew and Whitehead.[17 18] Trade union and stakeholder participants were identified from the research group's existing research users' network, using a snowballing technique. Potential participants were sent invitation letters, an outline of the research project and the objectives of the public involvement activity. Interested participants were given further information about the research and engagement activity through telephone conversations, face-to-face meetings and written correspondence.

> **Box 1  Description of material presented at the focused workshops**
>
> Participants were presented with a graph showing trends in Swedish employment rates of people with a limiting long-standing illness (LLI) and healthy people stratified by education level from 1990 to 2011, constructed from the Swedish Survey on Living Conditions and a blank timeline for the same period. Participants were then invited to chart important events or trends during this period on the timeline and to interpret developments in the employment of people with LLI and healthy people in the light of these events. Participants then compared graphs presenting differences in the employment rate of those with and without LLI and with high and low educational levels in Sweden, Denmark, the Netherlands and the UK for the period 2005–2011 from European Union Statistics of Income and Living Conditions data (EU SILC, EUROSTAT). They commented and reflected on the situation in Sweden compared with the other three countries. The differential rate of 'poverty risk' (<60% of median income) among non-employed/employed people with LLI and employed healthy people was also presented in graphical form (EU SILC), and participants were asked to interpret changes in poverty risk in light of the key sociopolitical or economic events they had charted on the timeline. Lastly, participants considered the categorisation of the UK, Netherlands and Denmark in relation to their degree of employment protection and economic security (flexicurity) in a 2×2 matrix and reflected where to best position Sweden and the trajectory of the country's position over time.

The planning, process and outcomes of the focused workshops were recorded and evidenced in parallel to the activity. Preliminary research results were presented to participants in graphical form during focused workshops, and then participants' perspectives on selected research findings were elicited.

Two researchers with expertise in qualitative research methods and public engagement, who were independent from the primary study, designed and led the public involvement activity and conducted the analysis. An independent moderator facilitated the workshops.

### Description of focused workshop activity

Selected findings were presented at four 2-hour focused workshops held separately with each public and stakeholder group at their workplace during June and August 2014. Two to six participants attended each session, with a total of 17 participants. Findings were presented in diagrammatic form (as described in box 1). One researcher moderated the discussion, and one or two members of the core research team presented the findings. A standardised topic guide was followed (online supplementary appendix 1), with additional questions tailored to suit each group's expertise and interest. Focused workshops were conducted and transcribed in Swedish, and selected quotations were translated into English.

### Analytic framework

Full transcripts of the digital recordings of the focused workshops were coded and categorised, employing the principles of content analysis, for the dataset as a whole and separately for each workshop.[19 20] Thematic analysis

**Table 1** Summary of explanations for epidemiological results from participants of focused workshops

| What explains lower employment rate in target group? | Which policy changes have occurred? | What explains increasing rates of relative poverty among non-employed? | Has Sweden moved to flexicurity? |
|---|---|---|---|
| Globalisation: higher demands on workers, higher education requirements. Economic recessions in early 1990s, early 2000 and 2008. (TU, E, EA and P) | Fighting inflation instead of unemployment, to be competitive (TU). Deregulation of many sectors (eg, transport and construction). (TU) | Increasing market incomes, tax reductions. (P) Stagnant levels of unemployment and social assistance benefits. (P) | Varies between individuals—if you are part of unemployment insurance and collective agreements, you are secure; otherwise not. Labour market not flexible. (E) |
| Changing labour market: New complex technology – increased demand on workers. Reduced demand for unqualified workers. (TU, E, EA and P) | Less emphasis on vocational training and life-long learning. (TU and EA) | Security has moved to other arenas: collective agreements and individual choices more important. (TU and E) | Flexibility for employers has increased. Security has become differentiated, not general. Depends on where you are in the system. (TU) |
| Changing composition of group with low education and LLI—more low educated immigrants with refugee background. (E and EA) | Social security reform in 2008: reduced duration of sickness absence, restrictions on disability pension. (TU, E, EA and P) | Fewer eligible for disability pension, fewer covered by unemployment insurance (EA). | Decreased economic security, not increased flexibility for the employer. Varies between different employers and white-collar/blue-collar trade unions, depends on which sector. (EA) |
| Increasing levels of mental ill health—more difficult to find jobs, lock-in effects. Focus on individual's efforts. (EA) | Reduced economic security to incentivise work. (EA) | Reduced economic security to incentivise work. (EA) | Politicians divided on the issue—most agree that flexibility is high; economic security is also considered high, but social democrats and green party think security has been lowered. (P) |
| Social insurance reform shifted large groups from social insurance to unemployment. (TU and EA) | Security has moved to other arenas—collective agreements and individual choices more important. (TU and E) | Increased unemployment insurance fee—lower participation. (TU and P) | |
| Target group less likely to be employed. (E) | Extended period for employers to pay sickness benefits. (E) | More unemployed rely on social assistance. (EA and TU) | |

EA, employment agency; E, employers; LLI, limiting long-standing illness; P, politicians; TU, trade unions.

of the moderator's notes and the transcripts drew out key themes from each workshop to form a summary of themes across the workshops and to compare the insights of the four different public and stakeholder groups between workshops.[21] Transcription and analysis were undertaken alongside the focused workshop activities as part of an iterative process that informed each phase of data collection.[22] The Swedish and UK research teams as a whole then reflected on the findings from the focused workshops and considered how these informed the researchers' original interpretation of their epidemiological data or generated new research questions for future research.

### Research ethics

Ethical clearance was not required under Swedish law on ethical review (refer to Lag 2003:460), as the information collected was not considered to constitute sensitive personal information (§3 in the Law). Stakeholders participated as collective members of their employing institution, not as individuals or the subjects of the research. All gave informed consent to being interviewed and for the interviews to be recorded and transcribed.

### RESULTS

Participants comprised employed men and women aged 40–60 years (17 participants). All groups commented on the decline in the employment rate of low educated people with LLI over time and offered a range of explanations for this development (table 1).

### Explanations for the lower employment rate of low educated people with LLI in Sweden

Participants put forward the economic recessions of the early 1990s, early 2000s and 2008, as explanations for the low employment rate of low educated people with LLI in Sweden:

> That's how it is. We know that in every recession […] they fall further and further down. They are last in line. (Trade unionist)

Many considered economic cycles to have had a greater influence on employment rates than national policies. The employers' association and some of the politicians emphasised that the pressures of global competition had led to outsourcing of low-skilled production tasks abroad, placed greater exigencies on workers in employment

and removed demand for low-skilled workers with health problems. As one politician put it:

> … we have a labour market which does not have room for these people; as [was the case] … in the 1990s … (Politician)

The 2008 reform of Swedish social insurance limited the time allowed for sickness absence and shifted a large group of people from social insurance to unemployment benefits. This was noted by the employment agency:

> [T]hose expelled from the insurance… the first batch … - [I] think it was 4th January 2010 - … 14,000 people suddenly came to the employment agency.

The composition and size of the low educated group with LLI was also seen to have changed. The group had become smaller, as a result of the rise in the educational level of the general population and included a higher proportion of immigrants and people with mental ill-health. This was seen as another explanation for the group's lower employment rate:

> A large proportion of recent immigrants are very low educated or illiterate and of course they have greater difficulties [in securing employment] … (Employment agency).

Another employment agency employee said that it was harder for people with functional disabilities resulting from mental illness to find jobs:

> … it is easier to adjust physically … but if you have ADHD and forget your appointment time …

A change in workplace attitude towards people with disabilities was also mentioned as a factor contributing to the difficulty for low educated people with LLI to find employment:

> The climate has sort of become tougher. (Employment agency)

### What are the key policy changes in Sweden that have influenced the employment of people with low education and LLI from 1990 to 2010?

Trade unionists suggested that changes in economic policy underpinned some of the trends in the employment rate of low educated people with activity limitations due to health:

> I believe that the higher structural unemployment that results from prioritizing low inflation over low unemployment enables employers to pick and choose. And then they pick the high educated … rather than the low educated.

Deregulation of many different sectors was also mentioned:

> [The] taxi [industry] in Sweden is the most de-regulated in the world.

Trade unionists felt that there had previously been much more emphasis on providing vocational training and continuing education for people with LLI who were not employed, whereas today individuals—particularly blue-collar workers—are simply urged to look for and obtain a job:

> … you would think our government could step in and make sure these groups get at least an equal chance and support blue collar workers.

Some participants, including trade union representatives, described how universal social security systems had been eroded over time. Unemployment insurance contributions had increased substantially, resulting in many employees leaving the scheme. Unemployment benefit levels had stagnated or been lowered, and social assistance benefits had plateaued.

In parallel with the erosion of universal social security systems, collective agreements between employers and employees had changed, partly to compensate for this phenomenon. As one trade unionist put it:

> Security has moved to other arenas, because, for instance, if you don't increase the unemployment insurance, it has moved to other types of insurance, that's how it is, then parts of the security system have moved.

The 'other arenas' included collective agreements. During the discussion, the trade unionists explained that up to the 1990s, Sweden used to have national policies and security systems that would cater for all in the same way, in case of unemployment or redundancy. The active labour market policies (eg, retraining) benefited those with lower education. The unemployment insurance was quite generous and adequate (at least for those with low/normal incomes). However, over the years, as the benefit levels were not increased and active labour market policies were reduced, the different unions tried instead to supplement with their own insurances, linked to specific collective agreements. The white-collar unions with lower unemployment risk could negotiate better terms than blue-collar unions, and as a result, the differential increased further and became dependent on the unemployment risk. Those in the most marginal positions on the labour market (and not subscribing to the unemployment insurance) were even worse off.

Employers mentioned another policy influence on people with LLI: an extension to the period of time an employer is responsible for paying sickness benefits for an absent employee, thereby increasing the cost to the employer. This policy change may have been detrimental to the employment chances of persons with LLI. Other reforms to social security regulations, including the aforementioned restriction to the duration of sickness absence and also to eligibility for disability pension, were considered by many participants to be important policy changes aimed at incentivising work:

It's tougher to be an outsider [to the labour market] today than [it was] 6 years ago … This is partly due to a political desire to strengthen the incentive to work. (Employment agency)

### What explains increasing rates of relative poverty among non-employed persons with LLI in different European countries?

Workshop participants were shown results comparing the prevalence of relative poverty among employed and non-employed people with LLI in Sweden, Denmark, the Netherlands and the UK, for the years 2005 and 2010, respectively. In Sweden, the poverty rate had nearly doubled from 2005 to 2010 although from a very low starting point, whereas increases in the other countries, which started with higher poverty rates in 2005, were smaller. Some politicians questioned the validity of the measure 'relative poverty' (defined as having an income below 60% of the median national income):

… it describes an increasing income difference in society, but it says nothing about whether poverty has increased. These are two completely different things.

Other participants attributed the increasing rates of relative poverty directly to national policies:

… it has been a deliberate policy … to increase the differences in income according to whether you work or do not work, … so you notice the change if you start working. (Employment agency)

The social insurance reform of 2008 had excluded many non-employed people with LLI from sickness benefit, yet many were not on the unemployment register. The issue was how they made their living:

… they are somehow supported by relatives … The group with mental problems … seek social assistance more than others. (Employment agency)

The increase in unemployment insurance contributions (due to a change in government policy in 2007) led to many employees leaving the insurance and was also put forward as an explanation for rising rates of relative poverty among this population group:

… there are many unemployed people who are not part of the unemployment insurance scheme at all, and who don't get anything… . (Employment agency)

### Has Sweden moved to flexicurity?

Workshop participants were asked whether they considered that the policy changes that had taken place from 1990 to 2010 had moved Sweden to qualify as a 'flexicure' country and were asked to place Sweden in a matrix of high/low flexibility and high/low economic security (figure 1). (Denmark, the Netherlands and the UK were positioned by the researchers prior to the workshops in line with the evidence from the policy literature.) Opinions were divided. One trade unionist commented:

The whole of [Swedish] working life has slipped towards the UK [position, with high flexibility/deregulation and low economic security] …

Another trade unionist also thought that flexibility had increased:

In Sweden, we have perhaps the highest flexibility out of many countries, to fire people … It is incredibly liberal … We belong to high flexibility.

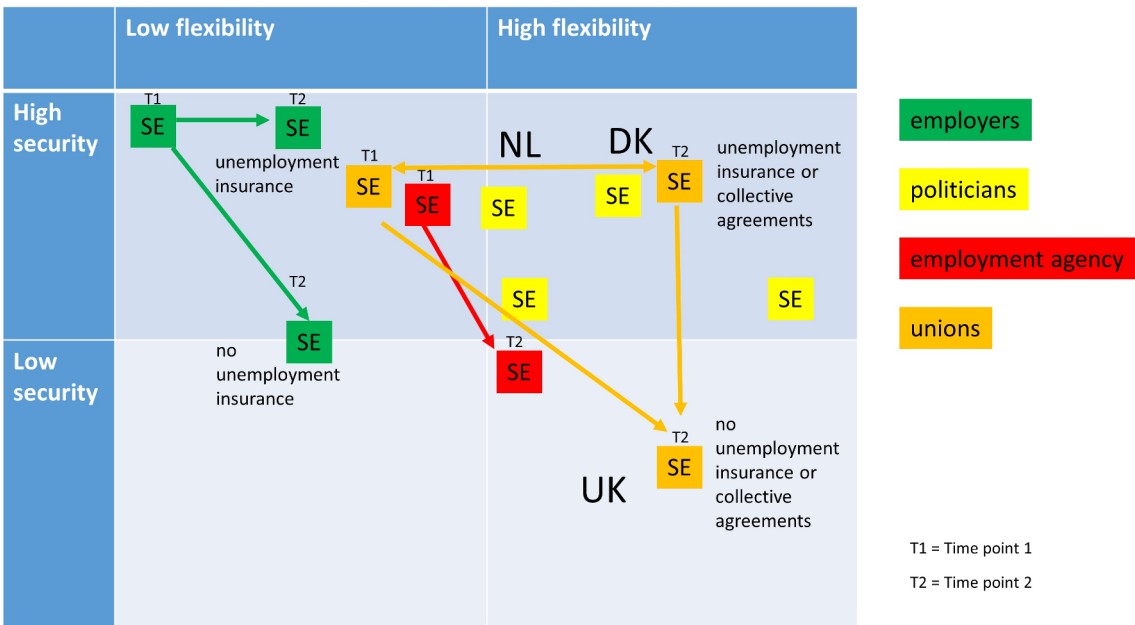

**Figure 1** Matrix showing the different categorisations of Sweden's degree of flexicurity by the four public groups. DK, Denmark; NL, The Netherlands, SE, Sweden.

Employers, however, considered that flexibility was low but that the degree of economic security differed according to whether an employee was part of unemployment insurance and covered by collective agreements. For those who did not belong to these schemes, security was lower than before:

> … the universal welfare system has been hollowed out since the 1990s. (Employer)

Trade unionists and employers referred to a two-tier system, in which the situation for 'insiders' to the labour market—those who have had a job for some time, who are part of the unemployment insurance scheme and are covered by collective agreements—was much better than for those outside these arrangements. Most politicians commented that Sweden had high flexibility and high security, having moved from lower flexibility and even higher economic security. One politician spoke of:

> … a divided labour market … There are those who have security and those who do not …

### Impact of this involvement on researchers' interpretation of findings

Subsequent to the focused workshops with public and stakeholder representatives, the wider research team comprising Swedish and UK members held reflective sessions to consider if, and how, the involvement activity had influenced their initial interpretation of findings or highlighted gaps in the evidence base.

Four main insights were identified, triggered by the public and stakeholder involvement, which deepened our understanding of our findings and influenced subsequent publications.

First, was the extent to which people experiencing the labour market and welfare reforms first hand (at 'the coal face' so to speak) sensed more drastic and nuanced change in the Swedish welfare system over the past two decades than was evident from the academic literature reviews and policy analyses. The exercise resulting in figure 1 highlighted this point visually.

Second, was the introduction of the concept of 'insiders' and 'outsiders' in the labour market, emerging from the workshops, particularly from discussions among trade unionists and employers. This highlighted the differing experiences and consequences for employees depending on socioeconomic status (SES) or simple blue-collar/white-collar dichotomy and on the length of time employees had had a job and whether they were part of the whole infrastructure that provided financial and job security. This concept fed into the generation of hypotheses to be tested in secondary data analysis, the results of which we subsequently published in a peer-reviewed paper.[15]

Third, was the possibility of differential effect of collective agreements for different occupational groups. It was the deep understanding by unions and employers of how collective agreements had operated in practice over the years that offered the researchers fresh insights into how groups that were already disadvantaged could be hardest hit by poorer, or no, collective agreements. This led to the development of a research proposal on the role of collective agreements in contributing to differential impacts of labour market policies, which was submitted by the research team to a Swedish Research Counsil: FORTE.

Fourth, triangulation of perspectives from the different public and stakeholder groups gave a much fuller picture of what was happening as a result of the interactions of people with the system. The trade unions and employers raised issues about the consequences of collective agreements as detailed above. The employment agency described from first-hand experience the impact of changing policies on the lives and finances of unemployed people, particularly those who were 'outsiders'. The politicians understood, from their own experience of legislating the stated and real reasons for changes in policies and the intended and unintended consequences.

## DISCUSSION

This is a rare example of a study exploring what difference public and stakeholder involvement makes to an evaluation of a natural policy experiment to influence the wider social determinants of health. Furthermore, it was conducted in a European country outside the UK, where the concept of lay involvement is less developed. To our knowledge, this is also the first report of involving the public and stakeholders in population-wide epidemiological studies, which require interpretation of national quantitative datasets.[4]

Reservations have been expressed elsewhere about the difficulties of involving the public in research that is highly technical.[23] Added to that was the population-wide nature of the policies being evaluated. Our decisions on *who* to involve and *how* were therefore critical to the whole research endeavour reported here. Our final selection of trade union representatives from blue-collar and white-collar unions as the most appropriate members of the relevant public, coupled with the decision to involve stakeholders from the employers' organisation, public employment office and politicians serving on employment policy committees, proved both feasible and productive. Participants in the focused workshops demonstrated a high level of lay expertise and a capacity to engage with the intricacies of policy developments, changes to the economy, the country's demographic profile and global market forces, from both a practical and a theoretical perspective. They expressed appreciation of the way the study findings were presented to them for comment—as '*big trends over time*'— rather than facing them, for example, with the technicalities of the epidemiological analysis. There is continuing debate about how to facilitate lay involvement in research and at what point would be most appropriate.[24–26] The focused workshops reported here provide an example of what helped public and stakeholder involvement in the

context of an evaluation of a population-wide natural policy experiment.

The study concluded that this public and stakeholder involvement benefited the interpretation of the research findings and conceptualisation of future research in a number of ways. Participants sensed more drastic and nuanced change in the Swedish welfare system over recent decades than was evident from our literature reviews and policy analysis. They also elaborated hidden developments in the Swedish labour market that were increasingly leading to 'insiders' and 'outsiders', with differing experiences and consequences for financial and job security. The explanation by the trade unionists and employers of the differential effects of the various collective agreements for different occupational groups was new and had not been a focus in the studies performed by the researchers. The first-hand experience of the employment agency provided new insights into how changes to the social protection system were contributing to the increasing trends in poverty among unemployed people with LLI. The politicians provided further reasoning behind some of the policy changes and their intended and unintended consequences. These insights fed into our subsequent reporting of the results of the flexicurity evaluation,[15 27 28] as well as the conceptualisation of new research that could be pursued in a future programme (see below).

### Limitations

The main limitation was that public and stakeholder involvement was limited to one discrete phase of the research process: the interpretation of initial findings. There is some evidence to suggest that involving the public throughout the research process (as well as long-term involvement and continuity of membership beyond a single research project) enhances impact.[4 29 30] While we judged involvement in this discrete phase to be the most practical and logical starting point faced with the specific population-wide policy evaluation of flexicurity, it became clear during the focused workshops that this could also be a way of building a continued relationship for future research. The research ideas coming out of the workshops and the willingness on the part of participants to be involved in a future programme laid the groundwork for involvement in future research from the very beginning, including the conceptualisation of the research questions.

A higher level limitation is that research on public and stakeholder involvement is inherently context specific, as the impact of involvement is likely to depend on many factors, including the topic being studied, the public groups being represented, the perspective of the researchers and the extent of involvement. Producing generalisable evidence in the area of public involvement is inherently difficult. This study was conducted in the context of Sweden and its labour market and social protection system. Mindful of this, we tried to tailor the public and stakeholder involvement initiatives to take account of the country context (Sweden), the topic (policies around flexicurity) and the nature of the research in which involvement is being encouraged (population-wide epidemiological studies). In this type of research, understanding and taking account of the context is essential if the findings are to be of value to research users in Sweden and elsewhere.

### Input into the conceptualisation stage of future research

The public and stakeholder involvement, although only at the interpretation of findings stage for the current research, gave rise to new research questions to address in a future research programme. Could it therefore be a way of introducing public involvement right from the conceptualisation stage in future population-wide epidemiological studies of this nature? Ideas for future research that came out of the public and stakeholder involvement included the following.

First, was the question of how the changing composition of, and the increasing prevalence of neuropsychiatric diseases in, the group of low educated people with LLI might affect the employment chances of the group. This is an interesting issue to pursue in view of the increasing demand for social skills on the labour market including in lower skilled jobs.

Another idea for future studies came from the unionists, regarding the differential conditions associated with different collective agreements, stating that 'security has moved to other arenas', that is, from the national level to the level of specific agreements negotiated by individual unions. Here, detailed assessment of conditions pertaining to different groups of manual workers, compared with conditions among higher non-manual employees, could help identify differential impacts and trajectories of different groups in the event of unemployment. This idea has been worked up, jointly with participants, into a research proposal, which was submitted to a Swedish Research Counsil: FORTE.

Third, the employment agency offered the use of their more detailed data to study the employment and economic situation of specific groups, such as people with psychological illnesses, whose employment prospects and economic situation may be poorer than among those with physical illnesses, not least following recent restrictions in social insurance policies.

Hence, early contacts with the public and with stakeholders, who day to day face the concrete impact of new policies or other changes on the labour market, may be very pertinent to specifying research questions addressing the impact of such real life changes. Increased involvement and dialogue with the public and with stakeholders in the identification and framing of specific research questions may therefore contribute both to increased scientific rigour, as well as societal relevance of policy-oriented research.

### CONCLUSION

This study shows that it is possible to incorporate public and stakeholder involvement in a meaningful way in epidemiological public health research. The insights and reflections the public and stakeholders offered added depth and

subtlety to the researchers' interpretation of the research findings and highlighted the issues perceived to be of most public relevance. In addition, participants raised questions that could not be answered by the current study, which stimulated ideas for future research and contributed to specifying highly relevant research questions. Early engagement with the public and with stakeholders may therefore contribute to improve both the scientific rigour and the relevance of policy-oriented research.

**Contributors** MW and BB conceived the research study. LN and RMAdC designed and led primary data collection and analysed the data. RMAdC and BB wrote the first draft and together with MW and LN interpreted the data. All authors contributed to the final draft.

**Funding** This research was funded by a European Commission FP7 grant DEMETRIQ (developing methodologies to reduce inequalities in the determinants of health) (grant reference: 278511), http://www.demetriq.eu/.

**Disclaimer** The funders had no role in study design, data collection and analysis, decision to publish, or preparation of the manuscript. The authors of this paper alone are responsible for the views expressed in this publication which do not necessarily represent the decisions or policies of the EU or their institutions.

**Competing interests** None declared.

**Patient consent** Not required.

**Provenance and peer review** Not commissioned; externally peer reviewed.

**Data sharing statement** Transcripts of the focused discussions, field notes and observations are in Swedish. To protect the anonymity of the public and stakeholder participants, we cannot share the primary qualitative data with third parties.

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
