## [Reviewer comments · BMJ Open]

ARTICLE DETAILS

TITLE (PROVISIONAL)	Impact of public involvement in interpreting research findings: the effect of labour market policies on inequalities in employment and health outcomes
AUTHORS	Anderson de Cuevas, Rachel Nylén, Lotta Burström, Bo Whitehead, Margaret

VERSION 1 - REVIEW

REVIEWER	Chris Patterson
REVIEW RETURNED	07-Nov-2017

GENERAL COMMENTS	Thank you for the opportunity to review this interesting manuscript on a valuable topic. I think the manuscript has some substantial issues that prevent me from recommending it for publication in its present form, but I think that revising the manuscript to address these issues would be worthwhile as producing evidence about the role of public involvement (PI) in the research process is potentially very valuable. My main issue with the manuscript is a lack of focus that makes it confusing and does not satisfy me that the research was robust. This lack of focus potentially results from having two research questions that are quite different from each other. On the one hand, the manuscript is about reporting various public groups' perspectives on flexicurity policies - this is described and reported clearly and effectively. On the other hand, the manuscript is about evaluating the impact of the PI process on the research process, and on researchers themselves, and I feel that this second research aim is underserved by the manuscript. While the findings related to public stakeholders' perspectives are clear and thorough, the findings related to researchers' experiences are written as if a summary of key points, rather than a thorough reporting and analysis of research data. For example, while the authors claim to have interviewed the researchers before and after the PI activity, the insights from the before-and-after data are not communicated effectively in the findings. As another example, the section entitled "Research context and attitudes towards public involvement" section contributes very little insight into either researchers' or participants' attitudes towards public involvement. The focus on the perspectives of the public representatives over the the perspectives of researchers is also evident in the Study design and methods section; there is good description of the recruitment of the public group and the workshops that were conducted, but insufficient description of the conduct or analysis of the interviews
---

with researchers. The methods section mentions "the comments of academic peers" (as a distinct source of data from researchers' perspectives), but doesn't explain who those peers were, how they were recruited, and how their comments were collected and analysed. Similarly, there is a focus group mentioned on Page 7 that is never described in terms of participation, a topic guide, analysis etc.. These limitations are such that the research is not replicable given the information in the manuscript.

The discussion is well written and reasonable, but might benefit from more critical examination of the limitations of the research, including limitations that are inherent to researching PI. The limitations section notes that individuals with activity limitations due to health problems are not involved, but perhaps this limitation warrants more consideration. Throughout the manuscript I question the use of 'public' to describe these groups. While the groups do represent certain types of publics, intuitively the term public feels that it should refer to the people ultimately affected by the issues being studied, rather than the experts and people in power who are involved in making decisions that affect those publics. Perhaps using a term like "expert groups" or "stakeholder groups" would be more clear. However, if the authors are certain that their use of "public" is typical of this area of research, I would be satisfied with its use. A higher-level limitation worth observing is that this research is inherently context-specific, as the impact of PI on a research process is likely to depend on so many factors: the topic being studied; the public groups represented; the attitudes and perspectives of the researchers; and the extent of the involvement of the public groups. Producing generalisable evidence in this area is inherently difficult.

As well as greater consideration of the limitations of the research, I would welcome greater consideration of the significance of the research findings. While it is positive that the PI enhanced the research in this case, might there be situations in which PI could undermine or derail researchers? Given the stark differences in the perspectives of the different groups involved in the workshops, it seems that the composition of panels of public representatives is essential, as a non-representative group with could introduce bias, particularly where they have vested interests.

For me to be satisfied with this manuscript, there would need to be a great deal of clear, replicable detail added about the research methods, particularly those relating to researchers, rather than the PI workshops. I recognise that doing this satisfactorily may result in an unusually long and complex manuscript, in which case the research may be better presented as two manuscripts: the first reporting the findings from the workshops, and the second reporting on the influence of PI on the research process. The inconsistent methodological detail about how the second research aim was investigated in the present manuscript suggests that the impact of the PI on the research process may have been studied in a relatively unstructured, ad hoc way, which would be understandable given the complex, real-world conditions in which the data were gathered. If this is the case, the findings simply may not be sufficiently robust for a research paper.

Thanks again for the opportunity to review this research. I have included miscellaneous minor comments below.

	Miscellaneous points: Sample: It would be useful to provide details of how many of the 17 public participants were in each group (employers, politicians etc.) Abstract: The objectives described here seem slightly different to those described in the main body of the paper. The project didn't exactly "represent the concerns of groups in weak positions in the labour market" as those groups weren't represented in the research sample. Rather, it represented the perspectives of various expert stakeholder groups. Abstract: The results section only reports results from stakeholder workshops - no results related to researchers. Therefore, only the first research aim is addressed. Page 8 Line 43 - "...after Petticrew and Whitehead" would read better as "using a process developed by Petticrew and Whitehead" Page 10 Line 13 - It isn't clear to me what "for the data set as a whole and separately for each workshop" means. In what way was content analysis applied at the level of individual workshops as well as across the whole dataset? Page 11 Line 7 - "age" should be changed to "aged" Page 14 Line 10 - "attitude" should be changed to "attitudes" Page 18 Figure 1 - What do the lines and arrows between boxes mean? Clearer explanation of this would be helpful.
--	--

REVIEWER	Dr Jonathan Boote
REVIEW RETURNED	10-Nov-2017

GENERAL COMMENTS	Although this is an interesting paper, I do not think it will be of sufficient interest to readers of this journal to merit publication in BMJ Open. I recommend that the authors consider journals more focused on labour market economics
---

REVIEWER	Kath Maguire
REVIEW RETURNED	06-Dec-2017

GENERAL COMMENTS	This is a well written paper. As the authors identify, there is limited literature on both methods and impact of involvement in quantitative analysis, making this also a welcome and potentially very useful paper. It could, however be significantly strengthened. In my view it is a mistake for the authors to frame this paper in terms of 'public' rather than 'stakeholder group' involvement. Using the term 'members of the public' to describe stakeholders who "participated as collective members of their employing institution" (p10:42) is a hostage to fortune and likely to detract from the reception of this paper among public involvement activists and researchers. It is to be welcomed that the research team describes this as the beginning of their engagement with a range of stakeholders beyond academia, and it is particularly welcome that they have identified
--

that it would also be useful to include members of the groups who are the subject of their research (people with limiting longstanding illness) in future work. However the term 'public involvement' in the context of public health research implies a broader citizen engagement than has been achieved here so it would be sensible to avoid the term.

Whether framing this as stakeholder or public involvement it might be useful to make reference to GRIPP2

(<http://www.bmj.com/content/358/bmj.j3453>)

It was confusing to me that the methodology section described stakeholder involvement as workshop participation with the researchers being interviewed, while in the results sections members of the stakeholder groups were sometimes described as 'interviewees'.

Given that the paper is framed in terms of the impact involvement has had on the interpretation of result findings it would be stronger if the differences between the views of workshop participants and the initial interpretations of the research team were made more explicit in the results section. These sections, and both Table 1 and figure 1 as well, often left me asking 'but how was this different from what researchers thought?' Mostly these sections reiterated stakeholder perspectives and differences between stakeholders. The researchers' analysis is mentioned on page 21 (5-13) but the quote then added is from a trade unionist – did the researchers' analyses pre or postdate the trade unionist's observation? How does this relate to impact of involvement?

Here (and in the sections on: influence of involvement; context and attitudes; and impact on the research process) I was looking for material from the interviews with researchers to demonstrate differences made, but this was sparse. The only quotation from a researcher seemed to be an expression of surprise at the view of policy makers (21:20-24), but not whether/how that changed analysis.

I was confused by the statement "difference of opinion between the stakeholder groups in interpreting Flexicurity, made the researchers realise that the policy is conceived differently within and between countries and the experience of different actors also varies." (p 21:45-50) as it seems to imply the involvement of stakeholders from more than one country – or does it mean people have difficulty interpreting the experience of other countries?

It is also quite weak to talk about 'additional material for reflection and analysis' (p22: 43-5) without outlining more specifically what new understandings have been elicited. The points about the value that building relationships with stakeholders which follows and that involvement raises questions for future research are good ones. The conclusion is, however, very weak and lacks real insight into the impact on interpretation suggested by the title.

All in all I think the authors need to clarify their main argument. If what they want to do is report the impact of the views of these stakeholder groups on researchers this needs to be clarified and foregrounded. The methodological aspects of this paper are useful and could be presented in themselves, but it would also be useful to have a clear presentation of the impact involving stakeholders can have on analysis and on developing future research directions.

VERSION 1 – AUTHOR RESPONSE

Reviewer(s)' Comments to Author:

Reviewer: 1: Chris Patterson, University of Glasgow

Reviewer's comment: My main issue with the manuscript is a lack of focus.... This lack of focus potentially results from having two research questions that are quite different from each other. On the one hand, the manuscript is about reporting various public groups' perspectives on flexicurity policies - this is described and reported clearly and effectively. On the other hand, the manuscript is about evaluating the impact of the PI process on the research process, and on researchers themselves, and I feel that this second research aim is underserved by the manuscript... the research may be better presented as two manuscripts: the first reporting the findings from the workshops, and the second reporting on the influence of PI on the research process [and the second may not be sufficiently robust for a research paper.

Authors' response: We agree fully with the reviewer's critique of the reporting of the second research question. We have therefore taken up the reviewer's suggestion of concentrating the revised manuscript on the first research question – reporting various groups' perspectives on the findings on flexicurity policies - which the reviewer judges to be reported clearly and effectively. The manuscript has been edited throughout the methods, results and discussion sections, to remove the reporting of the second research question.

Reviewer's comment: The discussion is well written and reasonable, but might benefit from more critical examination of the limitations of the research, including limitations that are inherent to researching PI. The limitations section notes that individuals with activity limitations due to health problems are not involved, but perhaps this limitation warrants more consideration. Throughout the manuscript I question the use of 'public' to describe these groups. While the groups do represent certain types of publics, intuitively the term public feels that it should refer to the people ultimately affected by the issues being studied, rather than the experts and people in power who are involved in making decisions that affect those publics. Perhaps using a term like "expert groups" or "stakeholder groups" would be more clear. However, if the authors are certain that their use of "public" is typical of this area of research, I would be satisfied with its use.

Authors' response: We thought long and hard about "who is 'the public?'" in the context of the evaluation of population-wide employment policies. We needed to avoid 'tokenism', whereby a few low skilled workers or unemployed people with disabilities were selected at random, without consideration of the nature of the research data they would be engaging with. Following discussions with PPI experts and deep knowledge of the health inequalities literature, we decided that it would be more appropriate to involve trade union representatives from blue-collar and white collar unions as members of the relevant public. They were workers themselves, but also had the added experience of how their fellow workers in different socio-economic circumstances and with different health conditions fared under the various national policies. They were also familiar with considering labour market trends at the population level. We are confident in referring to the trade unionists' involvement as 'public involvement' in the specific context of this population-wide public health study. We accept, though, that the other groups involved – from the employers' organisation, the state employment office and politicians - are more appropriately considered to be "stakeholder groups", as the reviewer suggests. We have therefore, revised the text throughout, to refer to "public and stakeholder involvement" and added a detailed explanation at the beginning of the methods section on the criteria we used to select the most appropriate 'public' and 'stakeholder' groups (all new text in red print). We believe that this could now make a contribution to the theoretical debate on 'Who is the public?' in population-wide epidemiological studies.

Reviewer's comment: A higher-level limitation worth observing is that this research is inherently context-specific, as the impact of PI on a research process is likely to depend on so many factors: the topic being studied; the public groups represented; the attitudes and perspectives of the researchers; and the extent of the involvement of the public groups. Producing generalisable evidence in this area is inherently difficult.

Authors' response: a discussion of this limitation has now been added the limitations section of the Discussion.

Reviewer's comment: As well as greater consideration of the limitations of the research, I would welcome greater consideration of the significance of the research findings. While it is positive that the PI enhanced the research in this case, might there be situations in which PI could undermine or derail researchers?

Authors' response: It is a fair point that PI could undermine or derail researchers, and this would have been relevant had we still been reporting on the second research question. But, as explained above, we have removed the reporting of that second research question from the revised manuscript, so we are no longer considering impact on researchers themselves in this paper. We therefore did not see the need to raise it as an issue in the revised manuscript, especially given the limited space available and the additions we need to make.

Reviewers' Miscellaneous points: Sample: It would be useful to provide details of how many of the 17 public participants were in each group (employers, politicians etc.). Authors' response: these details now added.

Abstract: The objectives described here seem slightly different to those described in the main body of the paper. The project didn't exactly "represent the concerns of groups in weak positions in the labour market" as those groups weren't represented in the research sample. Rather, it represented the perspectives of various expert stakeholder groups. Abstract: The results section only reports results from stakeholder workshops - no results related to researchers. Therefore, only the first research aim is addressed. Authors' response: abstract has been completely re-written in the light of the removal of the reporting of the second research question, and clarification on public and stakeholder involvement as above. .

Page 8 Line 43 - "...after Petticrew and Whitehead" would read better as "using a process developed by Petticrew and Whitehead". Authors' response: corrected.

Page 10 Line 13 - It isn't clear to me what "for the data set as a whole and separately for each workshop" means. In what way was content analysis applied at the level of individual workshops as well as across the whole dataset? Authors' response: removed as relates to second research question.

Page 11 Line 7 - "age" should be changed to "aged": Authors' response: corrected.

Page 14 Line 10 - "attitude" should be changed to "attitudes". Authors' response: corrected.

Page 18 Figure 1 - What do the lines and arrows between boxes mean? Clearer explanation of this would be helpful. Authors' response: figure 1 re-drawn and explanation added.

Reviewer: 2: Dr Jonathan Boote: University of Sheffield

Reviewer's comment: Although this is an interesting paper, I do not think it will be of sufficient interest to readers of this journal to merit publication in BMJ Open. I recommend that the authors consider journals more focused on labour market economics.

Authors' response: this paper is about public and stakeholder involvement in public health research. It is highly relevant to readers of BMJ Open. It would be of very little interest to readers of journals focussed on labour market economics.

Reviewer: 3: Kath Maguire, University of Exeter Medical School, UK

Reviewer's comment: This is a well written paper. As the authors identify, there is limited literature on both methods and impact of involvement in quantitative analysis, making this also a welcome and potentially very useful paper. It could, however be significantly strengthened. In my view it is a mistake for the authors to frame this paper in terms of 'public' rather than 'stakeholder group' involvement. Using the term 'members of the public' to describe stakeholders who "participated as collective members of their employing institution" (p10:42) is a hostage to fortune and likely to detract from the reception of this paper among public involvement activists and researchers.

It is to be welcomed that the research team describes this as the beginning of their engagement with a range of stakeholders beyond academia, and it is particularly welcome that they have identified that it would also be useful to include members of the groups who are the subject of their research (people with limiting longstanding illness) in future work. However the term 'public involvement' in the context

of public health research implies a broader citizen engagement than has been achieved here so it would be sensible to avoid the term.

Authors' response: Thank you for this important observation which was also made by Reviewer 1. We have clarified what we mean by 'public' in the context of population-wide epidemiological research and also introduced the concept of 'stakeholder involvement'. Hopefully, the text is no longer a hostage to fortune and may even add to the theoretical debates in this field. We thought long and hard about "who is 'the public?'" in the context of the evaluation of population-wide employment policies. We needed to avoid 'tokenism', whereby a few low skilled workers or unemployed people with disabilities were selected at random, without consideration of the nature of the research data they would be engaging with. Following discussions with PPI experts and deep knowledge of the health inequalities literature, we decided that it would be more appropriate to involve trade union representatives from blue-collar and white collar unions as members of the relevant public. They were workers themselves, but also had the added experience of how their fellow workers in different socio-economic circumstances and with different health conditions fared under the various national policies. They were also familiar with considering labour market trends at the population level. We are confident in referring to the trade unionists' involvement as 'public involvement' in the specific context of this population-wide public health study. We accept, though, that the other groups involved – from the employers' organisation, the state employment office and politicians - are more appropriately considered to be "stakeholder groups", as the reviewer suggests. We have therefore, revised the text throughout, to refer to "public and stakeholder involvement" where appropriate and added a detailed explanation at the beginning of the methods section on the criteria we used to select the most appropriate 'public' and 'stakeholder' groups (all new text in red print).

Reviewer's comment:

It was confusing to me that the methodology section described stakeholder involvement as workshop participation with the researchers being interviewed, while in the results sections members of the stakeholder groups were sometimes described as 'interviewees'. Given that the paper is framed in terms of the impact involvement has had on the interpretation of result findings it would be stronger if the differences between the views of workshop participants and the initial interpretations of the research team were made more explicit in the results section. Here (and in the sections on: influence of involvement; context and attitudes; and impact on the research process) I was looking for material from the interviews with researchers to demonstrate differences made, but this was sparse.

Authors' response: We accept fully this criticism that the reporting of the impact on researchers is confusing and not as rigorous as the reporting of the impact of stakeholders' involvement on interpretation of findings. Reviewer 1 made a similar comment, and we have responded to both by taking up Reviewer 1's suggestion of concentrating the revised manuscript solely on the impact of the stakeholders' involvement on interpretation of research findings and on developing future research directions (as Reviewer 3 suggests). The manuscript has been edited throughout the methods, results and discussion sections, to remove the reporting of the impact on researchers themselves. In doing so, we have removed the sentences that reviewer 3 flagged up as inconsistent or confusing.

Reviewer's comment:

The methodological aspects of this paper are useful and could be presented in themselves, but it would also be useful to have a clear presentation of the impact involving stakeholders can have on analysis and on developing future research directions.

Authors' response: A new section at the end of the Results has been added which summarises the impact involving stakeholders had on the interpretation of findings and also on developing future research directions (new text in red print). The scope to influence future research directions is elaborated on in a new section "Input into the conceptualisation of future research" in the Discussion section (also indicated by red print).

Reviewer's comment: All in all I think the authors need to clarify their main argument:

Authors' response: we thank the reviewers for all their perceptive comments and suggestions. These have been very thought-provoking and have stimulated us to look with fresh eyes at the manuscript and where it is unclear and unfocussed. We now believe that the extensive revisions we have made in response as detailed above (all indicated in red print on the revised manuscript) have made this a much more tightly focussed and better paper.

REVIEWER	Chris Patterson
REVIEW RETURNED	23-Jan-2018

GENERAL COMMENTS	I thank the authors for their thorough response to my comments, and I think the revised manuscript is greatly improved. The newly-included rationale for choosing the research participants is persuasive, and overall the research meets the research aims and makes a good case for how this specific type of PPI can enhance research. Some minor observations:  - Perhaps Prof. Burström's address should read "Social Medicine" rather than "Social Medicin", as the rest of the address is in English. However, I may be incorrect as I know nothing about Swedish addresses. - In the abstract, the phrase "stakeholder involvement makes" could be changed to the past-tense "stakeholder involvement made", as this research reports on the impact on a past project, rather than ongoing impact. - There is a typo in the quote beginning "Security has moved to other arenas..." – a double comma is used.
--

REVIEWER	Kath Maguire
REVIEW RETURNED	22-Jan-2018

GENERAL COMMENTS	The authors have thoroughly revised the paper and the result is much clearer and more focused. I believe it makes an important and useful contribution to the literature on public involvement in public health research.
---

VERSION 2 – AUTHOR RESPONSE

Reviewer(s)' Comments to Author:

Reviewer: 3

Reviewer Name: Kath Maguire

Institution and Country: University of Exeter Medical School, UK

Please state any competing interests: None

The authors have thoroughly revised the paper and the result is much clearer and more focused. I believe it makes an important and useful contribution to the literature on public involvement in public health research.

Reviewer: 1

Reviewer Name: Chris Patterson

Institution and Country: University of Glasgow, United Kingdom

Please state any competing interests: None declared

I thank the authors for their thorough response to my comments, and I think the revised manuscript is greatly improved. The newly-included rationale for choosing the research participants is persuasive, and overall the research meets the research aims and makes a good case for how this specific type of PPI can enhance research.

Some minor observations:

- Perhaps Prof. Burström's address should read "Social Medicine" rather than "Social Medicin", as the rest of the address is in English. However, I may be incorrect as I know nothing about Swedish addresses.

Thank you. I agree this was confusing. I have amended the address to a Swedish address for consistency in line with the information against the affiliated institutions which precedes it. We have changed Karolinska Institute to Karolinska Institutet and retained Social Medicin in Swedish.

- In the abstract, the phrase "stakeholder involvement makes" could be changed to the past-tense "stakeholder involvement made", as this research reports on the impact on a past project, rather than ongoing impact.

We agree that the tense was not consistent and have changed this to 'made'.

- There is a typo in the quote beginning "Security has moved to other arenas..." – a double comma is used.

This has been changed to a single comma.